# Mechanical Strength and Thermal Properties of Cement Concrete Containing Waste Materials as Substitutes for Fine Aggregate

**DOI:** 10.3390/ma15248832

**Published:** 2022-12-10

**Authors:** Paweł Łukowski, Elżbieta Horszczaruk, Cyprian Seul, Jarosław Strzałkowski

**Affiliations:** 1Department of Building Materials Engineering, Faculty of Civil Engineering, Warsaw University of Technology, 00-637 Warsaw, Poland; 2Department of Civil and Environmental Engineering, West Pomeranian University of Technology in Szczecin, al. Piastów 50a, 70-311 Szczecin, Poland

**Keywords:** concrete, crushed bricks, crushed concrete, recycled aggregate, sewage sludge

## Abstract

The increasing volume of waste and the requirements of sustainable development are the reasons for the research on new waste management concepts. The research results presented in this paper show the effect of recycled aggregate on the selected properties of cement concrete. The aggregates obtained from three types of wastes are tested: recycled concrete paving, crushed ceramic bricks, and burnt sewage sludges. The recycled aggregates replaced 25% and 50% of the volume of the fine aggregate. The tested aggregates worsen the concrete mixes’ consistency and decrease, to some extent, the compressive strength of the concrete. However, the tensile splitting strength of the concrete with recycled aggregates is similar to that of the reference concrete. Using recycled aggregates worsens the tightness of the concrete, which manifests itself by increasing water penetration depth. The thermal properties of concrete are slightly affected by the type and content of the recycled aggregate. Considering the expected improvement in recycled aggregate processing, they can be an alternative to natural aggregates. Using recycled aggregates in cement concrete requires extensive studies to search for ways to increase their possible content without worsening concrete performance.

## 1. Introduction

The volume of produced industrial and vegetable wastes is continuously rising. The increasing pressure on activity in line with the sustainable development rules leads to research on using various industrial wastes in the construction and building materials industry [1]. Construction produces many wastes during the building structures’ demolition. Concrete, the main construction material, is manufactured from natural aggregates, cement and water, which makes it a relatively cheap and easy-to-produce material. Ordinary concrete contains about 75–80% aggregate. Therefore, numerous studies are recently focused on limiting the use of natural aggregate in concrete by replacing it with various industrial wastes [2,3]. Construction utilizes mainly industrial by-products. These wastes are usually stored in dumps. Such materials as silica fumes, fly ashes, and blast-furnace slag are used in construction on an industrial scale.

The separate group of industrial wastes are the sludges, which are the materials remaining from the industrial sewage or their treatment process. The term sludge can also concern the precipitated suspension formed during conventional water purification and many other industrial processes [4]. The wastes can be contaminated by toxic substances [5]. An increasingly common solution is burning the sewage sludges, which causes sanitization and reduces their volume. However, the burning method generates another type of waste, which can also be used in construction. Besides the increased content of heavy metals, the ashes have a chemical composition similar to the Portland cement clinker. They also show some pozzolanic activity, which suggests their potential usefulness as cement additives [6,7].

The other type of waste is the construction waste generated when constructing a new structure or renovating or demolishing an existing building. Most of the wastes mentioned above are such materials as timber, plasterboards, building ceramics, reinforcing steel, concrete, plastic, glass, and cardboard. Waste concrete is the basic raw material for producing recycled aggregate [8,9].

Agricultural wastes form another category. They are biodegradable with time, but they must first be stored in particular places (composting plants, silos). For utilization, they are often burnt, and the created powder can be used as a fine aggregate or filler for concrete and mortar [10,11,12].

All the above wastes can be utilized for manufacturing cement composites. However, only some are used on an industrial scale, particularly in cement production. The increasing volume of waste and the requirements of sustainable development are the reasons for the research on new waste management concepts. The conducted studies should make it possible to utilize the various wastes as raw materials in the building materials industry [13]. Concerning cement composites, most investigations are aimed at utilizing the wastes as an aggregate replacement, which is mostly justified from an economic point of view. However, the possibilities of using wastes as substitutes for the aggregate in cement concrete or mortar, without significantly worsening their performance, usually oscillate around 10% of the total aggregate mass. Such a low degree of substitution is not profitable in most industrial production cases [14]. The cleaning and preparation of the waste material (crushing, grinding, heating, etc.) generate substantial costs, often not considered within scientific research.

In Japan, the USA, and most EU countries, using aggregates made of construction wastes for cement concrete is standardized [9,15,16]. Considering the economics of the production process, concrete with processed waste aggregates can be an alternative to natural aggregate concrete if the lower cost of the recycled aggregate at least compensates for the necessary concrete strength correction costs. In Poland, recycled aggregates are successfully used in road construction [17]. Utilizing recycled aggregates in cement concrete production is less common. One reason is the lack of practical guidelines for such use [18]. Therefore, investigations of the various waste materials used for cement composite manufacturing are fundamental.

The recycled aggregates utilized as a substitute for the fine aggregate in concrete have high porosity, which means high water absorbability and weak interfacial transition zone between aggregate grains and cement matrix [19,20,21]. The consequence is a decrease in compressive strength [22,23]. The fine aggregate from concrete recycling is usually limited to non-structural applications, such as filling material for soil stabilization, geosynthetic structures, or substrates in road construction [24,25]. The porosity and water absorption of the fine aggregate from wall recycling, containing ceramic bricks, mortar, and plaster, is even higher [26,27,28,29]. Therefore, it is used more often in cement mortars than concrete [30,31,32,33]. The aggregate made from organic or inorganic wastes requires additional tests for its usability. The specific surface area, particles’ morphology, and chemical composition of the recycled aggregates can affect the mechanical performance and durability of concrete. However, despite these limitations, the shortening of natural aggregate resources enforces the search for new material solutions.

The subject of the studies presented in the paper was determining the relation between the type of recycled aggregate and selected properties of the concrete mix and hardened concrete. The aggregates obtained from three types of wastes with the established composition were tested:The aggregate from recycling the concrete paving;The aggregate from crushing the wastes created by ceramic brick production;The aggregate from burning the sewage sludges in the municipal water treatment plant.

It was assumed that the recycled aggregates will replace 25% and 50% of the volume of the fine aggregate (river sand). The aggregate replacement was not accurate regarding grain size distribution, which differs from the previous studies reported in the literature discussed in this section. In the investigation described in this paper, the volume of the individual fractions of the recycled aggregates was selected to minimize the strength decrease. The sand with a grain size of 0–2 mm was substituted with the recycled aggregate with a grain size of 0.5–4 mm, which made it possible to obtain the concrete with high content of the recycled fine aggregate (up to 50%) without significant worsening of the mechanical performance.

The presented research covered determining the chemical composition of the recycled aggregates regarding the possibility of their use in concrete and the influence of these aggregates on the performance of the concrete. Determining the technological parameters of the concrete mix, the mechanical performance, thermal properties, and tightness of the hardened concrete made it possible to thoroughly assess the modified concrete’s usability. The sewage sludges are produced in large volumes and are challenging to utilize. Therefore, the possibility of their use as a substitute for the fine aggregate in cement concrete is important and worth evaluating, yet there is little research on this topic. However, the most promising results were achieved for aggregate made of ceramic bricks wastes.

## 2. Materials and Methods

### 2.1. Materials

Reference concrete specimens CR0 were prepared using Portland cement CEM I 42.5R with a specific density of 3.1 g/cm^3^, 0–2 mm natural sand, 2–8 mm and 8–16 mm gravel, and tap water. The grain size distribution of sand and gravel is presented in Figure 1.

Because recycled aggregate usually causes a worsening of concrete mix workability [34], the CR0 concrete mix consistency was designed as S5 according to the European Standard EN-12350-2 [35]. The polycarboxylic superplasticizer (SP), with a density of 1.06 g/cm^3^, was used for this aim. The content of the superplasticizer was 1.5% of the cement mass.

In the subsequent concrete mixes, 25% and 50% of the sand volume was replaced by:Aggregate from concrete paving recycling; the paving was initially crushed, and the fractions larger than 2 mm were sifted on the construction sieves (Figure 2a). The concrete mixes containing this aggregate are marked CC25 and CC50. The bulk density of this aggregate was 0.75 g/cm^3^ and after compaction 0.93 g/cm^3^;Aggregate from the wastes created during ceramic bricks production (Figure 2b). The fractions of 1–4 mm dominated in this aggregate. The concrete mixes containing this aggregate are marked CB25 and CB50. The bulk density of this aggregate was 1.12 g/cm^3^ and after compaction 1.21 g/cm^3^;Aggregate from Pomorzany Water Treatment Plant (Szczecin, Poland) in the form of slag (SS), created during thermal deactivation of the sewage sludges (Figure 2c). The slag is manufactured by drying in the contact dryer and burning in the moving grate boiler [36]. The material is porous with a significant content of open pores [37]; thus, it has low mechanical strength and high water absorbability, which was demonstrated by the worsening of the consistency of the tested concrete mixes. The slag fractions of 0.5–2 mm were selected for preparing concrete mixes. The concrete mixes containing this aggregate are marked CSS25 and CSS50. The bulk density of this aggregate was 0.54 g/cm^3^ and after compaction 0.64 g/cm^3^.

The images of the used recycled aggregates are presented in Figure 2.

The grain size distribution of the recycled aggregates is presented in Figure 3.

### 2.2. Preparation of Concrete Mixes and Concrete Specimens

Seven concrete mixes were designed. Designing the concrete mixes was based on the previous experience of the authors. It was assumed that the reference concrete mix should demonstrate high fluidity so that the worsening of its workability caused by the introduction of the recycled fine aggregate is as tiny as possible. The composition of the reference mix CR0 is presented in Table 1. The following concrete mixes contained three types of recycled aggregate as volumetric substitutes for the sand (FA). The content of recycled aggregates was 25% and 50% of the fine aggregate. The remaining components were not changed. The compositions of the tested concrete mixes are presented in Table 1.

After mixing the components in the laboratory mixer, the concrete mixes were tested. Then, the specimens for hardened concrete testing were prepared. The strength and thermal conductivity were determined on the cubic specimens of 100 mm × 100 mm × 100 mm. The specimens, after preparation, were stored in the molds for 24 h. After demolding, the specimens were placed in the container filled with water at 20 ± 2 °C until testing.

### 2.3. Test Methods

In the initial testing range, the recycled aggregates’ chemical analysis was performed using the Bruker Quantax EDS X-ray energy-dispersive spectrometer.

The consistency of the concrete mixes was determined by the slump method according to the European Standard EN 12350-2 [35]. The concrete mixes were tested immediately after mixing the components. The thermal conductivity of concrete was determined on the cubic specimens of 100 mm × 100 mm × 100 mm, described in Section 2.2. The concrete specimens were stored in dry laboratory conditions until their masses stabilized. After cutting the specimens in half, the tests were conducted on the exposed concrete cores. The non-stationary technique was employed for thermal conductivity measurement using the Isomet 2104 apparatus. All measurements were performed on the central cross-sections of the specimens. Eight measurements were performed for each concrete. The values of thermal conductivity, *λ*, volumetric specific heat, *c*_v_, and thermal diffusivity coefficient, *a*, were determined.

Compressive strength was determined after 7 and 28 days of curing. The tests were carried out according to the European Standard EN 12390-3 [38], using the Toni Technik strength machine (Berlin, Germany) with a maximum load of 5000 kN. The strength of each concrete was determined on six specimens and was measured with an accuracy of 0.1 MPa.

Tensile splitting strength was determined on the cubic specimens of 100 mm × 100 mm × 100 mm according to the European Standard EN 12390-5 [39], after 28 days of curing, on the wet specimens. The Toni Technik strength machine (Germany) with a maximum load of 100 kN was used. Six specimens were tested for each concrete.

Additionally, the apparent density of concrete in the dry state and depth of water penetration under pressure according to the European Standard EN 12390-8 [40] were tested. The tests were conducted on the cube specimens 100 mm × 100 mm × 100 mm. After demolding the specimens, the surfaces the water could penetrate were ground. The tests started after 28 days of curing. The water pressure of 0.5 MPa was applied and kept for 72 h. Then, the specimens were split in half in the strength machine perpendicular to the penetrated surface. After splitting, the range of water penetration was measured with an accuracy of 1 mm. Three specimens were tested for each concrete.

## 3. Results and Discussion

### 3.1. Chemical Composition of the Tested Aggregates

The EDS analysis was performed for the identification of the chemical elements in the recycled aggregates. No heavy metals nor chlorides, which would exclude the material as a substitute for the sand in concrete, were found in the aggregate from concrete recycling (Figure 4). The total content of sulfur was less than 0.1%, which is far below the threshold value of 1% according to the European Standard EN 12620 [41].

The chemical composition of the aggregate from ceramic bricks recycling is presented in Figure 5. No heavy metals nor sulfur were found.

Figure 6 presents the chemical composition of the aggregate from the burnt sewage sludge.

The investigations of the ashes and slags from the thermal conversion of the sewage sludges are focused on their potential application in the production of building materials, such as glass ceramics [42,43,44], bricks [44,45], and cement mortar and concrete [46,47,48]. These additives are used as direct substitutes for clay, cement, or sand [44,49,50,51,52]. The EDS analysis showed no chlorides in the tested specimens. In the case of burnt sewage sludges, other researchers also confirm the lack of chlorides [48,53]. According to the Polish regulations [54], ashes and slags created during the sewage sludge burning can be used for preparing the mixes for construction purposes, excluding, however, the use in buildings destined for the permanent residence of people or animals and the buildings with food processing. They also do not fulfill standard requirements for concrete and cement [55,56].

### 3.2. Consistency and Apparent Density of Concrete Mixes

The results of consistency testing are presented in Figure 7. The biggest worsening of consistency was caused by the aggregate from the sewage sludge; for the CSS50 mix, it was more than 50% in relation to the reference mix CR0. The reason for this consistency downfall is the high porosity of the slag from sewage sludge. In the case of the recycled aggregates made by crushing and grinding, the increased water demand is also caused by a high content of dust fractions. The slag is additionally crushed while mixing the dry components. The behavior of the aggregate from the crushed concrete was similar. However, plastic consistency was achieved for the concrete mixes CC and CB.

The available studies results [48,57] show that due to the porosity of the recycled aggregates, their use requires the particular procedure of adjusting the amount of water dosed to the concrete mix. Figure 8 presents the SEM image of the slag grains with open pores.

Obtaining a desirable consistency of the concrete mix requires that the total water covers the water necessary for the cement hydration and the additional water absorbed by the aggregate. The water absorbed by the aggregate grains is then released. It can help cement hydration in the interfacial transition zone (ITZ) between the aggregate and hardened cement paste, thus improving the ITZ properties, including strength. However, the excess water, not used during hydration, evaporates and can leave the micropores weakening the composite’s structure. Therefore, designing the concrete mix with recycled aggregates always needs special experimental verification.

The average apparent densities of the concrete mixes were: 2370 kg/m^3^ and 2340 kg/m^3^ for CC25 and CC50, respectively; 2380 kg/m^3^ and 2350 kg/m^3^ for CB25 and CB50, respectively; 2375 kg/m^3^ and 2335 kg/m^3^ for CSS25 and CSS50, respectively.

### 3.3. Mechanical Performance of Concrete

#### 3.3.1. Compressive Strength

The compressive strength testing results of concrete after 7 and 28 days of curing are presented in Table 2 and Figure 9.

Every concrete with recycled aggregate showed decreased compressive strength compared to the reference concrete RC0. Except for the concrete containing the aggregate from the crushed bricks; the concrete CB50, after 28 days of curing, demonstrated slightly higher compressive strength (rise by 3.6%). The concrete CB25 had slightly lower compressive strength after 7 days of curing (a decrease of 3.7%). Generally, the CB concrete’s compressive strength was similar to that of the reference concrete CR0 within the range of 5%. These results were caused by the grain size distribution of the crushed bricks. The recycled aggregate made of crushed bricks comprised 80% of the grains with a size of 1–4 mm. Such a grain size distribution made it possible to achieve compressive strength close to reference concrete despite lower density.

The recycled aggregate made of sewage sludge had a grain distribution similar to that of the aggregate from the crushed bricks. However, due to its porous structure and low crushing resistance [57], concrete with this aggregate demonstrated the lowest compressive strength. The compressive strength of the concrete CSS25 after 7 days of curing was 17.2% lower, and after 28 days of curing by 10% lower, compared to the reference concrete. The compressive strength of the concrete CSS50 was lower by 22.6% and 13% after 7 and 28 days of curing, respectively. As it was demonstrated in [48], the compressive strength of the cement concrete and mortar containing burnt sewage sludges rises with curing time and, after 56 or 90 days, can be close to that of the reference concrete.

In most cases, the aggregate made by grinding and crushing the concrete from demolition or ceramic bricks and the aggregate from sewage sludge ash shows higher water absorption and lower apparent density [9,19,26,27,28,29,57]. The majority of reported studies demonstrate the downfall of compressive strength with increasing recycled aggregate content [9,22,26,34]. The research presented in [58,59] shows that using recycled aggregate with a larger grain size could make it possible to obtain compressive strength close to the concrete with natural aggregate. However, a significant decrease in strength was observed when the recycled aggregate content was 50% or higher. In the investigation described in this paper, the content of the aggregate fraction below 0.5 mm was substantially limited. As a result, the 28-day compressive strength of the concrete with 50% of the aggregate from waste bricks (CB50) was even higher compared to the reference concrete CR0.

The studies on using sewage sludge (SS) in cement concrete are mainly focused on utilizing this waste as a partial substitute for cement [36,47,50,60]. In the case of using SS for replacing the fine aggregate, tests are usually performed on the cement mortars [48,49,61]. Few works dealing with using SS as a substitute for the fine aggregate in cement concrete confirm that compressive strength decreases with increasing content of the sewage sludge in the concrete. In [62], the accepted content of SS in the total fine aggregate is suggested as 25% when the strength downfall did not exceed 5%. The study presented in this paper shows that the 25% content of sewage sludge (CSS25) leads to a 10% downfall in compressive strength. The downfall of the compressive strength at 50% content of SS (CSS50) was even more significant (Figure 9). Therefore, from the compressive strength point of view, 25% of sewage sludge ash in the fine aggregate seems to be the limiting value.

#### 3.3.2. Tensile Strength

The tensile splitting strength testing results of concrete after 28 days of curing are presented in Figure 10.

The highest tensile strength was observed, as in the case of the compressive strength, for the concrete CB50. The concrete CC50 and CSS50 demonstrated significantly lower tensile strength than the reference concrete CR0 (17% and 15%, respectively). The tensile strength of the remaining composites was close to that of the concrete CR0. The good tensile strength is a consequence of the porous structure of the aggregate grains’ surface [63]. The research presented in [58,59] confirms that the small content of the aggregate from the recycled bricks can lead to improvement in the tensile strength due to the more developed surface of the recycled aggregate grains compared to the natural non-broken aggregates.

### 3.4. Depth of Water Penetration

The results of testing water penetration depth (WPD) are presented in Figure 11. Figure 12 presents fractured concrete specimens after testing WDP.

Using recycled aggregates obtained by crushing or grinding the wastes often worsens the concrete tightness. This worsening is caused by increased porosity of the aggregates after crushing. Such a situation has also been observed for the tested materials. The poorest tightness was observed for the composites containing the aggregate from sewage sludges, CSS25 and CSS50. Due to the WPD above 50 mm, the latter can only be used in the concrete structures exploited in the exposed class X0 or XC1. As described in Section 3.2, the burnt sewage sludge has a very porous structure with many open pores, leading to the poor tightness observed in the tests.

Limiting the sewage sludge content in the fine aggregate to 25% is appropriate because of the compressive strength, consistency, and tightness. The water penetration depth under pressure exceeded 50 mm for the concrete with 50% of sewage sludge (CSS50). The poor tightness of the concrete containing 50% and more SS has also been confirmed in [48].

### 3.5. Thermal Conductivity

Figure 13 presents the average values of the tested composites’ thermal conductivity coefficients, *λ*. The recycled aggregates slightly decreased the thermal conductivity proportionally to their volumetric content. No evident influence of the aggregate type on *λ* values was found.

Even more minor was the influence of the recycled aggregate on the specific heat (Figure 14). Only at 50% content of the recycled aggregate in the sand was a decrease in *c_v_* observed (concretes CC50, CB50, and CSS50).

The highest thermal diffusivity, *a*, has been registered for the reference concrete CR0, where the average value of *a* was 1.19 × 10^−6^ m^2^/s, which is higher by 10% than in the case of the concrete CSS50, having the lowest *a* value (Figure 15). The type and content of the recycled aggregate only slightly affected the thermal diffusivity of concrete.

## 4. Conclusions

The presented research results show the effect of recycled aggregate on the selected properties of cement concrete. The used aggregates were obtained from recycling concrete and ceramic bricks, and from burnt sewage sludges. All these aggregates were introduced into the concrete mix as partial substitutes for the fine aggregate. The following conclusions can be formulated:The presence of chlorides or heavy metals, which could exclude the recycled aggregate as a building material component, was not detected.The recycled aggregates worsen the concrete mixes’ consistency. The reason for this phenomenon is high porosity and, in the case of the aggregates made by crushing and grinding, high content of the dust fractions. The downfall of the consistency can be limited by the initial saturation of the recycled aggregate with water before mixing the dry components [48,63].The recycled aggregates, due to their porous structure, cause a decrease in the compressive strength of the concrete. The only exception was the concrete containing the crushed bricks, which, after 28 days, demonstrated slightly higher (by 3.6%) compressive strength compared to the reference concrete. The downfall of compressive strength is decreased with time, and after 56 or 90 days of curing, the compressive strength can be close to the reference concrete, as demonstrated in [48].The tensile splitting strength is the highest for concrete with crushed bricks as a recycled aggregate. Using the other tested aggregates led to a tensile strength similar to the reference concrete.Using the recycled aggregates worsens the tightness of the concrete, which manifests itself by increasing water penetration depth.The thermal properties of concrete are slightly affected by the type and content of the recycled aggregate. The thermal conductivity is slightly decreased, while the specific heat and thermal diffusivity are not significantly influenced.The obtained test results and the analysis of the available literature data show that considering the concrete mix’s and hardened concrete’s performance, the sewage sludge’s content in the fine aggregate should not exceed 25%.

The material modification of concrete requires, besides the control of the strength, the control of properties affecting the durability, such as tightness, frost resistance, and chemical resistance [64,65]. The introduction of recycled aggregate can worsen certain technical properties of the concrete, such as compressive strength and tightness. Nevertheless, adequately produced and controlled recycled aggregates can be an alternative to natural aggregates.

Despite numerous research works, utilizing recycled aggregate for cement concrete production still needs extensive studies. The volume of industrial wastes is growing; therefore, further investigations should mainly concern the limited contents of various types of recycled aggregates and possible ways to increase these contents.

## Figures and Tables

**Figure 1 materials-15-08832-f001:**
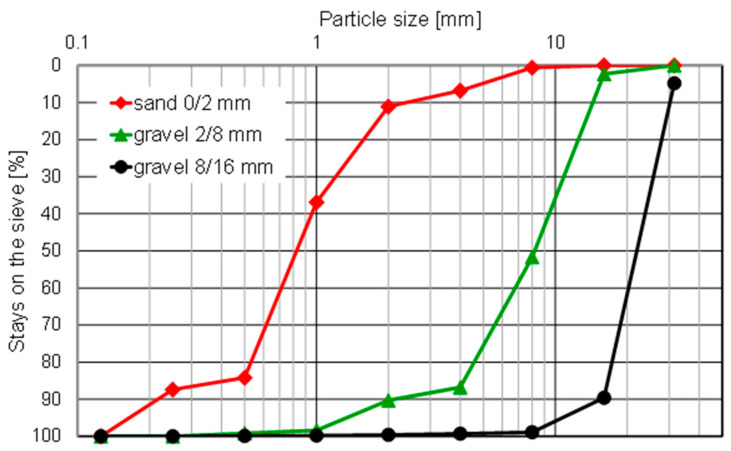
The grain size distribution curves of sand and gravel.

**Figure 2 materials-15-08832-f002:**
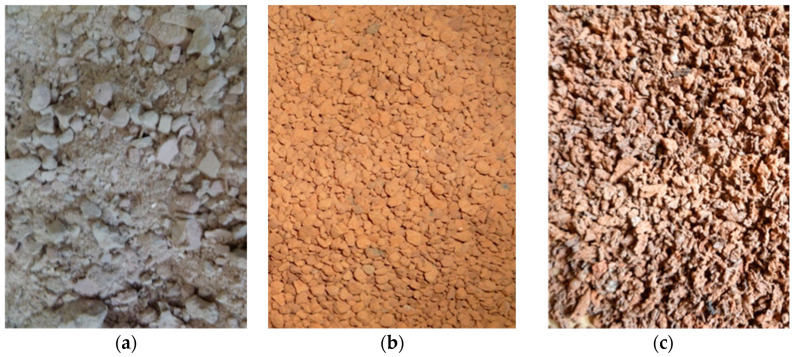
The recycled aggregates used in tests: (**a**) crushed concrete; (**b**) crushed ceramic brick; (**c**) burnt sewage sludges.

**Figure 3 materials-15-08832-f003:**
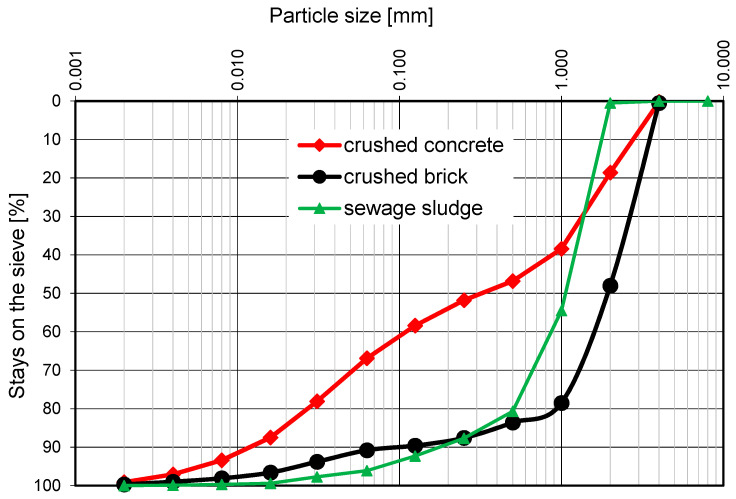
The grain size distribution curves of the recycled aggregates.

**Figure 4 materials-15-08832-f004:**
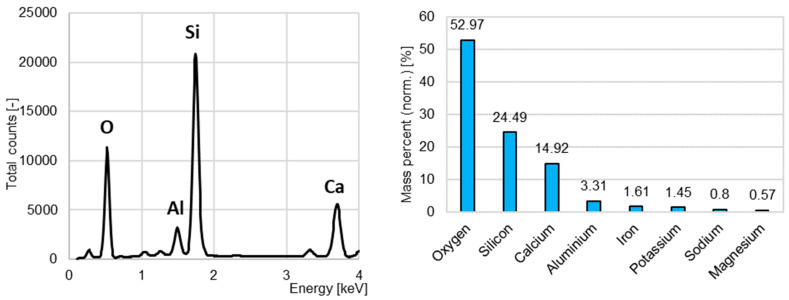
The energy-dispersive X-ray spectroscopy results for the concrete recycling aggregate.

**Figure 5 materials-15-08832-f005:**
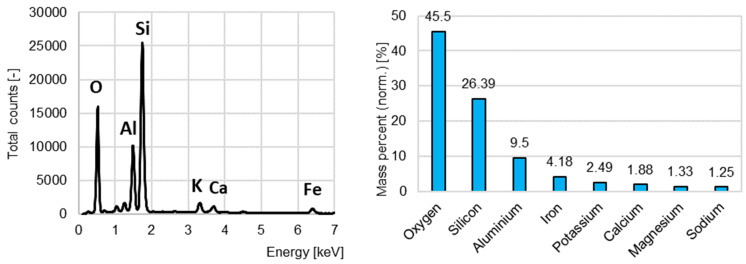
The energy-dispersive X-ray spectroscopy results for the ceramic brick recycling aggregate.

**Figure 6 materials-15-08832-f006:**
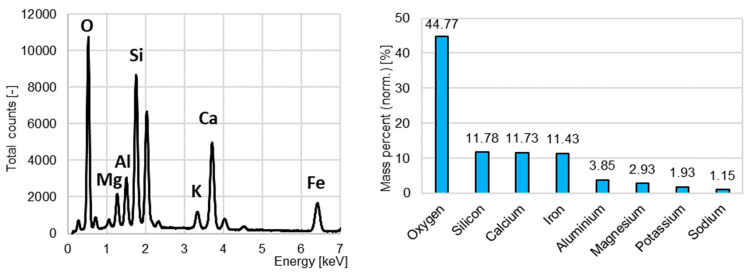
The energy-dispersive X-ray spectroscopy results for the burnt sewage sludge aggregate.

**Figure 7 materials-15-08832-f007:**
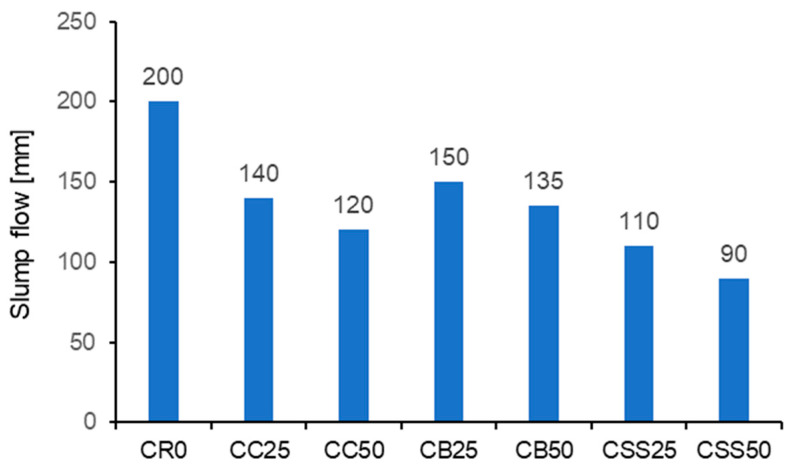
The results of the concrete mixes’ consistency testing.

**Figure 8 materials-15-08832-f008:**
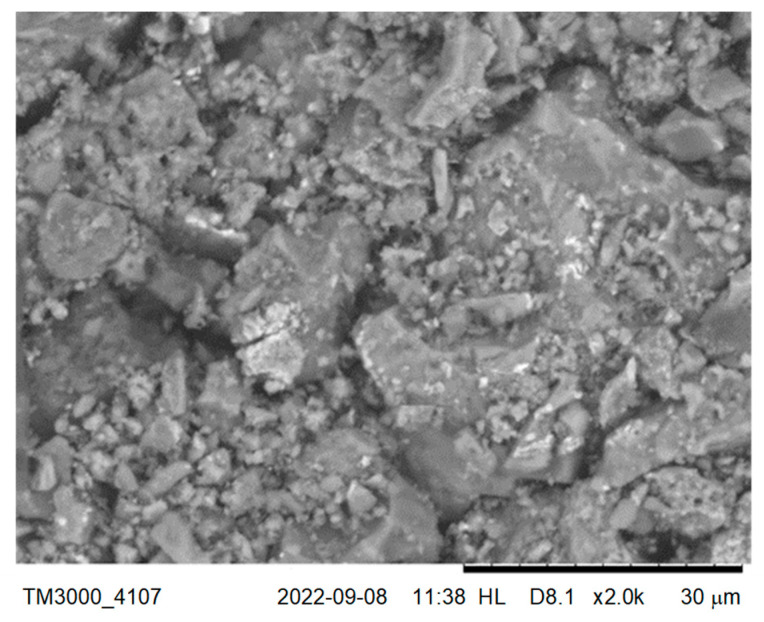
The SEM image of the slag from thermal conversion of sewage sludge.

**Figure 9 materials-15-08832-f009:**
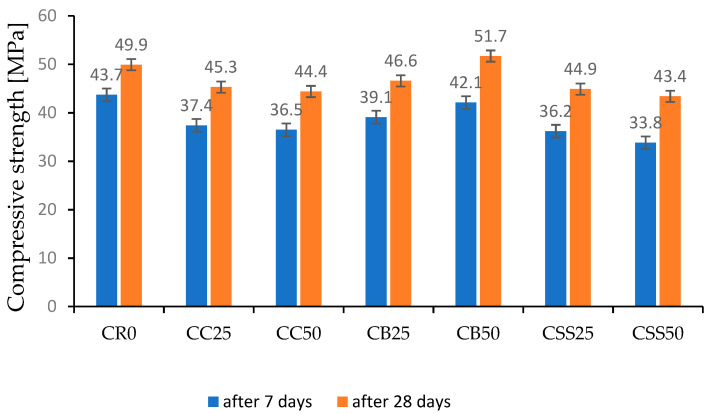
The test results of concrete compressive strength.

**Figure 10 materials-15-08832-f010:**
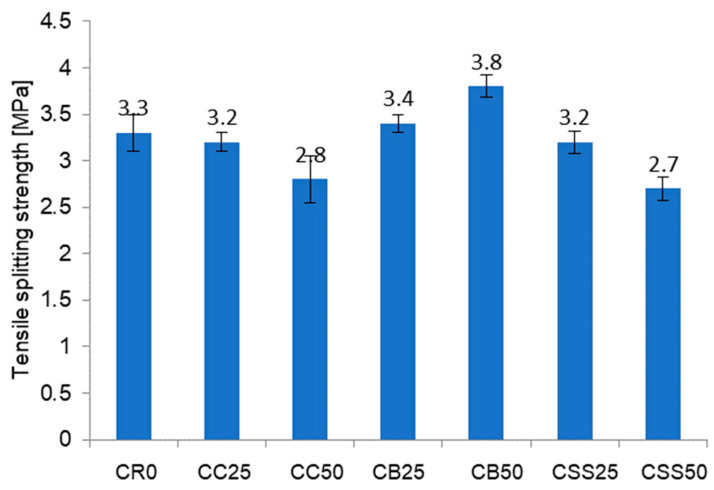
Test results of concrete tensile splitting strength after 28 days of curing.

**Figure 11 materials-15-08832-f011:**
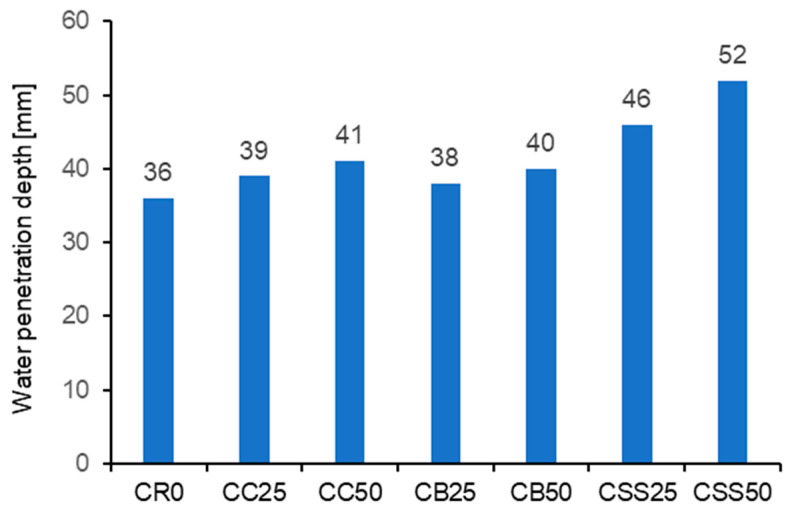
The results of testing maximum water penetration depth under pressure.

**Figure 12 materials-15-08832-f012:**
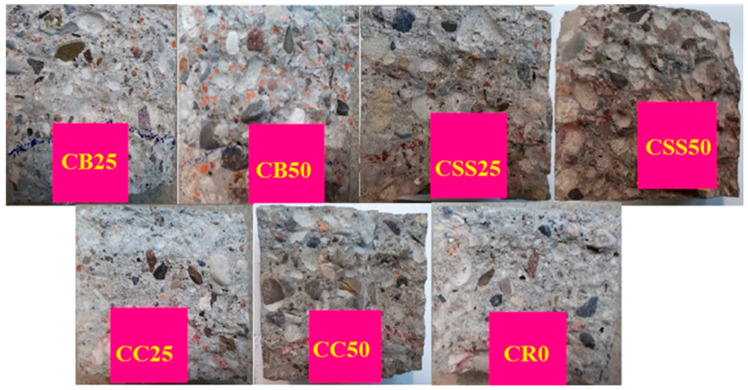
Concrete specimens after testing water penetration depth.

**Figure 13 materials-15-08832-f013:**
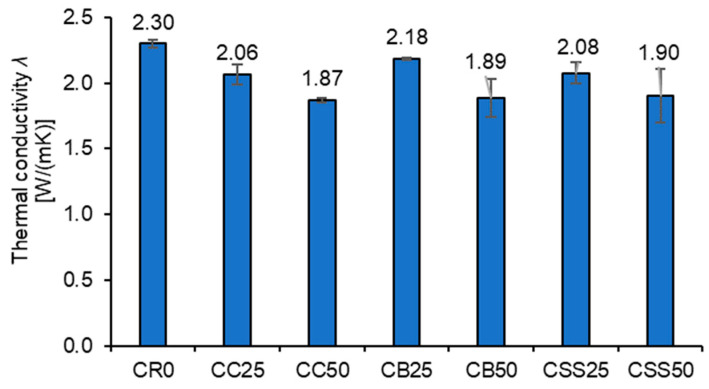
The average values of thermal conductivity coefficients of the tested composites.

**Figure 14 materials-15-08832-f014:**
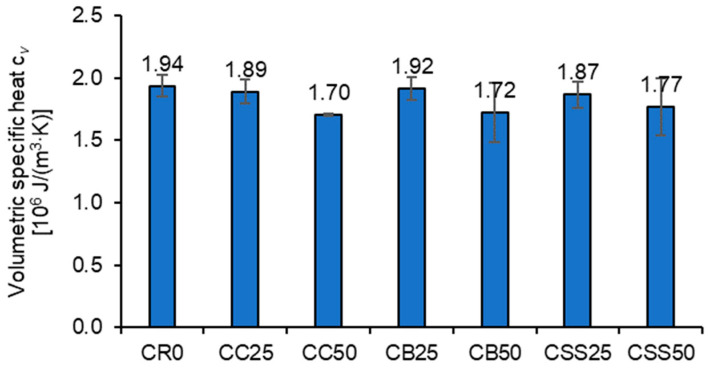
The average values of volumetric specific heat of the tested composites.

**Figure 15 materials-15-08832-f015:**
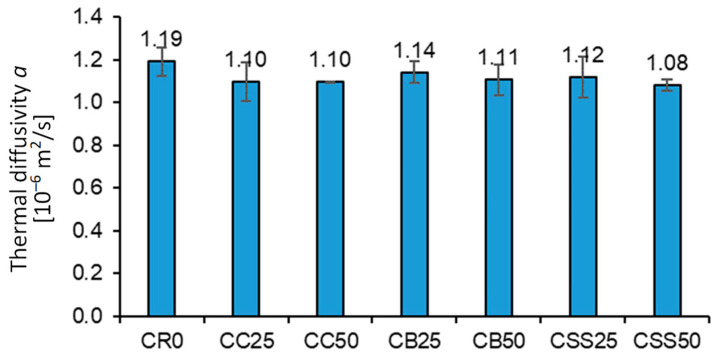
The average values of concrete thermal diffusivity.

**Table 1 materials-15-08832-t001:** The compositions of concrete mixes.

Concrete Designation	Cement	Water	Sand 0/2 mm	Crushed Concrete	Crushed Bricks	Sewage Sludge	Gravel 2/8 mm	Gravel 8/16 mm	SP
	[kg/m^3^]								
CR0	320	176	580	0	0	0	708	560	4.8
CC25	320	176	448	98	0	0	708	560	4.8
CC50	320	176	299	196	0	0	708	560	4.8
CB25	320	176	448	0	63	0	708	560	4.8
CB50	320	176	299	0	126	0	708	560	4.8
CSS25	320	176	448	0	0	36	708	560	4.8
CSS50	320	176	299	0	0	72	708	560	4.8

**Table 2 materials-15-08832-t002:** Test results of concrete compressive strength.

Concrete Designation	Compressive Strength [MPa]	
	After 7 Days	After 28 Days
CR0	43.7 ± 1.3	49.9 ± 1.8
CC25	37.4 ± 1.4	45.3 ± 2.3
CC50	36.5 ± 2.1	44.4 ± 1.9
CB25	39.1 ± 1.6	46.6 ± 1.4
CB50	42.1 ± 1.3	51.7 ± 1.8
CSS25	36.2 ± 1.6	44.9 ± 2.6
CCSS50	33.8 ± 1.8	43.4 ± 2.8

## Data Availability

The data presented in this study are available on request from the corresponding author.

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
