# Peer review of "Mechanical Strength and Thermal Properties of Cement Concrete Containing Waste Materials as Substitutes for Fine Aggregate"

_materials, 2022, doi:10.3390/ma15248832_

Round 1
Reviewer 1 Report
The paper is lacking novelty, since there are similar works in the literature. Therefore, I suggest rejection for this work. I attached my other comments for the paper.

Reviewer 2 Report
This manuscript reviewed the ‘Properties of cement concrete containing waste materials as substitutes for fine aggregate’. The manuscript is elaborately described and contextualized with the help of previous and present theoretical background and empirical research. All the references cited are relevant to this area of research. The conclusions are supported by the results. However, some corrections need to be addressed before the acceptance the Manuscript.
1. The present title is more general. It should be specific one for your research.
2. Include your research recommendation in the last line of abstract
3. Remove the keyword ‘properties’.
4. The sentence ‘The increasing pressure on activity in line with the sustainable development rules leads to research on using various industrial wastes in the construction and building materials 31 industry’ needs to be cited. Some related works are as follows.
https://doi.org/10.1002/suco.201800355
5. The sentence ‘Therefore, numerous studies are recently focused on limiting the use of natural aggregate in concrete by replacing it with various industrial wastes’ needs some more citations. Some works are listed below,
https://doi.org/10.1016/j.conbuildmat.2021.123209
6 How your study is different from existing studies?
7. Mention the novelty of your work.
8. It would be better if grain size distribution curve is drawn instead of table 1.
9. How the mix design was arrived?
10. Show the more experimental test photos.
11. Try to give better image for fig 7.
12. Draw the error bar for fig.8
13. Conclusion section is too lengthy.
Reviewer 3 Report
Paper is interesting. However, the title should highlight mechanical and thermal properties.
The literature review is limited and a lot of recent literature are available on waste material reinforcement in concrete. They should be adequately cited.
The specimen blocks should be shown in Figure. The crack images after compressive and splitting test should be shown as well.
Except Fig. 9, 11, 12, 13, the bar charts have no error bars. Please add them. Even in Fig. 13, there is no error bar for sample CC50. Why so?
In abstract and conclusion, the significant findings only should be mentioned. In the current version, the conclusion is too long.
Reviewer 4 Report
Reviewer1
Comments and Suggestions for Authors:
The authors have studied Properties of cement concrete containing waste materials as substitutes for fine aggregate. Even the work is appreciable, the paper lags in the following aspect:
Title
The title is insufficient to convey the purpose of your research efficiently.
Keywords
The keywords should not be too long. Also, the terms used should be sorted alphabetically.
Abstract
The abstract needs to be highly summarized by the author.
Introduction
The authors have not clearly mentioned any important limitations and recommendations.
The reviewed manuscript has practical relevance but I fail to see the novelty of the study as similar ones have been published multiple times in literature. The methodology in itself has nothing to add to the current state of art either. Authors are urged to stress how their study differs from previous ones and why it is novel.
Finally, discuss unanswered questions and potential future research. suggest for other researchers to do in their future works.
Materials and Methods
The experiment should introduce the main information of some important equipment.
Results and Discussion
Incorrect use (excess/lack) of articles (e.g. ‘a’, 'the')
Errors in the sentence structure. Authors need to improve the quality of language.
Conclusion
Conclusion needs to be more systematic by the author.
Round 2
Reviewer 1 Report
In my opinion, the authors have revised and improved the manuscript in accordance with the requirements, and I therefore suggest accepting it.
Reviewer 2 Report
Al the comments are addresses well.
Reviewer 3 Report
Can be accepted.
Reviewer 4 Report
The paper has been changed and can be accepted.